# Neighbourhood socioeconomic disadvantage and body size in Australia's capital cities: The contribution of obesogenic environments

**Suzanne J. Carroll**[ID]⊕*, **Michael J. Dale**[ID]⊕, **Gavin Turrell**⊕

Australian Geospatial Health Laboratory, Health Research Institute, University of Canberra, Canberra, Australian Capital City, Australia

⊕ These authors contributed equally to this work.
* suzanne.carroll@canberra.edu.au

## Abstract

Residents of socioeconomically disadvantaged neighbourhoods have higher rates of overweight and obesity and chronic disease than their counterparts from advantaged neighbourhoods. This study assessed whether associations between neighbourhood disadvantage and measured body mass index (BMI) and waist circumference, are accounted for by obesogenic environments (i.e., residential distance to the Central Business District [CBD], supermarket availability, access to walkable destinations). The study used 2017–18 National Health Survey data for working-aged adults (aged ≥18 years, n = 9,367) residing in 3,454 neighbourhoods across Australia's state and territory capital cities. In five of eight cities (i.e., Sydney, Melbourne, Brisbane, Adelaide, and Perth) residents of disadvantaged neighbourhoods had significantly higher BMI and a larger waist circumference than residents of more advantaged areas. There was no association between neighbourhood disadvantage and body size in Hobart, Darwin, and Canberra. Associations between neighbourhood disadvantage and body size were partially explained by neighbourhood differences in distance to the CBD but not supermarket availability or walkable amenities. The results of this study point to the role of urban design and city planning as mechanisms for addressing social and economic inequities in Australia's capital cities, and as solutions to this country's overweight and obesity epidemic and associated rising rates of chronic disease.

## Introduction

In Australia and many other developed countries, residents of socioeconomically disadvantaged neighbourhoods experience higher rates of morbidity and mortality for cardiovascular disease (CVD) and Type 2 diabetes (T2D) than residents of more advantaged neighbourhoods [1–3]. Neighbourhood inequities in anthropometric factors such as overweight and obesity likely contribute to associations between neighbourhood disadvantage, CVD and T2D. Overweight and obesity are strongly associated with these (and other) chronic degenerative

require applicants to undertake ABS directed training to ensure the maintenance of confidentiality. Application may be made via: www.abs.gov.au/websitedbs/D3310114.nsf/home/How+to+Apply+for+Microdata.

**Funding:** The author(s) received no specific funding for this work.

**Competing interests:** The authors have declared that no competing interests exist.

conditions [4–7], and studies conducted in Australia [8, 9], North America [10, 11], and elsewhere [12, 13] indicate that residents of disadvantaged neighbourhoods, particularly women, are more likely to be overweight or obese.

The emergence and fast rise of the obesity epidemic during the last few decades is often attributed to exposure to 'obesogenic' neighbourhood environments [14, 15], that is, environments with an urban design and built form that dissuade or inhibit physical activity, and/or a food environment that makes choosing healthy and nutritious food difficult or unaffordable. Studies examining associations between obesogenic environments and physical activity have reported that activity is higher in more walkable, more compact, and less sprawling neighbourhoods [16, 17]. Activity promoting neighbourhoods are typically residentially dense and located close to employment and walkable destinations (e.g., shops, recreation facilities, social infrastructure such as health care centres, and libraries), and these proximate built environment features facilitate increased physical activity, less reliance on private motor vehicles, greater use of public transport and active travel (e.g., walking and cycling), and shorter commutes. In terms of food and nutrition, obesogenic neighbourhood environments have limited availability or restricted access to shops such as supermarkets which sell a variety of food, including healthy options, and/or greater densities of take-away and fast-food stores selling nutrient deficient energy dense food [18]. Living in such neighbourhoods has been associated with poorer diets and nutrient intakes [19–21] and higher rates of overweight and obesity [22, 23].

Inequities in overweight and obesity between socioeconomically advantaged and disadvantaged neighbourhoods are seen to be a consequence of differential exposure to obesogenic environments. From this perspective, residents of disadvantaged neighbourhoods are more likely to be overweight and/or obese due to greater exposure to food-environments that hinder healthy eating, and/or environments that inhibit physical activity and promote sedentary behaviour [24–26]. To date, Australian studies examining these associations have produced mixed results. Some find that socioeconomically disadvantaged neighbourhoods are more likely to have built environments with obesogenic characteristics [27–29], others have reported no differences between advantaged and disadvantaged neighbourhoods [30, 31], and some studies show that disadvantaged neighbourhoods are more conducive to obesity-protective behaviours such as walking for transport [32] and physical activity in local parks [33]. Australian capital cities are generally conceived of as mono-centric, anchoring on a central business district (CBD) [34, 35]. These cities typically have low levels of suburban housing density [36] resulting in urban sprawl and peripheral or distal suburbs substantially distant from, and different to, inner city suburbs in terms of access to amenities, resulting in spatially patterned inequity [36]. Lower levels of liveability are likewise evident in distal suburbs in Australian capitals [37, 38]. It remains an open question whether socioeconomically disadvantaged neighbourhoods in Australia are obesogenic to the extent that their environmental characteristics shape physical activity and dietary behaviours, and whether these obesogenic environmental characteristics contribute to the higher rates of overweight and obesity in disadvantaged neighbourhoods.

This study aims to offer a partial answer to the foregoing question by documenting the association between neighbourhood disadvantage and body size and examining the contribution of obesogenic environments to this association. Specifically, we investigate whether differences in measured body mass index (BMI) and waist circumference between socioeconomically advantaged and disadvantaged neighbourhoods are accounted for by neighbourhood differences in residential distance to the CBD, the food environment, and access to walkable destinations. Previous international and Australian studies of obesogenic environments and overweight and obesity have primarily been conducted in one geographic context such as a

single city or region [18] and the untested assumption is that findings pertaining to a particular city or region are generalisable to other (different) cities or regions. In this study we test this assumption by performing separate analyses for each of Australia's eight state and territory capital cities. In so doing, we examine whether and to what extent relationships between neighbourhood disadvantage, obesogenic environments, and body size in any given capital city is generalisable to all other capital cities.

## Materials and methods

This cross-sectional observational study received ethics approval from the University of Canberra's Human Research Ethics Committee (UC HREC-2313) and has been conducted in compliance with relevant laws and institutional guidelines, and in accordance with the requirements of the Declaration of Helsinki.

The study uses data from the 2017–18 Australian National Health Survey (NHS) which was designed and implemented by the Australian Bureau of Statistics (ABS). The 2017–18 NHS data were enriched by the ABS using a Geospatial Information System to include contextual environmental information. Full details of the scope and coverage of the NHS, its research design, sampling procedures, and data collection methods have been documented elsewhere [39]. Only a brief overview is provided here.

### Sampling

The NHS was conducted over a 12-month period (July 2017 to June 2018) to take account of possible seasonal effects. Urban, rural, and remote areas in all six States and two Territories of Australia were sampled. A stratified multistage area sample of private dwellings (n = 25,109) was used, intending to provide detailed estimates for capital cities. After excluding out-of-scope dwellings (e.g., selected households which had no residents in scope of the survey, vacant or derelict buildings, buildings under construction) the number of sampled dwellings was reduced to 21,544, and of these, 16,384 dwellings responded to the survey (76.1% response rate). Within each responding dwelling, one adult aged 18 years and over, and one child 0–17 years (where applicable), were randomly selected, resulting in a total sample of 21,315 persons.

### Data collection

Trained ABS personnel collected data using face-to-face Computer Assisted Personal Interviews. The information collected included long-term health conditions, disability status, psychological health, medication use, health literacy, health-related behaviours and risk factors, and household sociodemographic characteristics. Indicators of each respondent's physical and social environment were also included as part of the NHS data set and those of relevance for the present study were neighbourhood socioeconomic disadvantage, count of supermarkets, and count of walkable amenities (each described in detail below). For our study, combinations of geographic indicators available in the NHS dataset were used to identify areas corresponding to each of the eight state and territory capital cities in Australia: Sydney, Melbourne, Brisbane, Adelaide, Perth, Hobart, Darwin, and Canberra.

### Directed Acyclic Graph (DAG)

We constructed a Directed Acyclic Graph (DAG) to make explicit our conceptualisation of the associations between neighbourhood disadvantage, residential distance to the CBD, and body size, and thus guide model specification (Fig 1). The pathways in the DAG inform our study in five ways. First, individual-level sociodemographic factors influence selection into

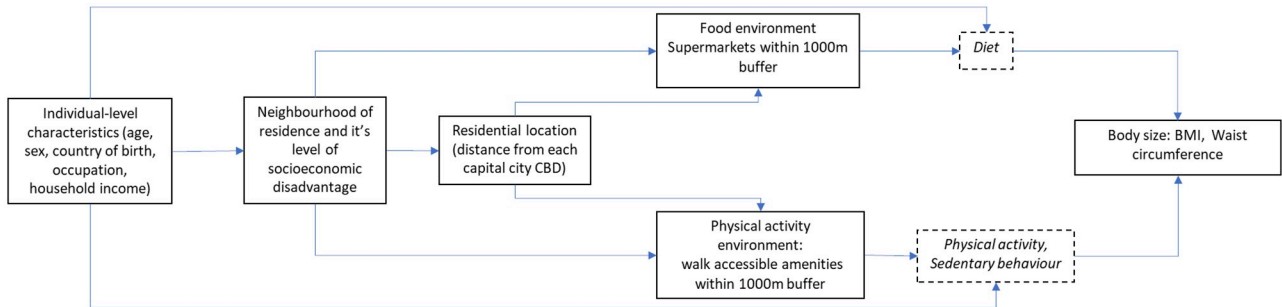

**Fig 1. Directed Acyclic Graph (DAG) depicting relationships between neighbourhood socioeconomic disadvantage, obesogenic environments, and body size (body mass index and waist circumference).** Variables included in analyses are denoted by boxes with solid outlines. Boxes with dashed lines and italic text represent untested variables and are included in the DAG for completeness.

neighbourhoods that differ in their socioeconomic characteristics. Second, neighbourhood socioeconomic disadvantage will be associated with residential distance to the CBD: disadvantaged neighbourhoods on average, will be located further away from the CBD than advantaged neighbourhoods. Third, advantaged and disadvantaged neighbourhoods, in part because of their differing proximity to the CBD, will differ in their counts of supermarkets and walkable amenities. Fourth, neighbourhood differences in counts of supermarkets and walkable amenities will shape physical activity and dietary behaviour and thus contribute to neighbourhood differences in BMI and waist circumference. Note that whilst diet, physical activity, and sedentary behaviour are depicted in the DAG as being proximal causes of body size, they are presented for completeness and to demonstrate the theoretical and biological plausibility of the DAG: they are not included in our model specification, as doing so would arguably constitute over-adjustment.

## Exposure measures

**Neighbourhood-level socioeconomic disadvantage.** Statistical Areas Level 1 (SA1s) in each capital city were assigned an area-level socioeconomic score using the ABS' Index of Relative Socioeconomic Disadvantage (IRSD) [40]. The IRSD scores were calculated using 2016 census data and derived by the ABS using Principal Components Analysis. An SA1's IRSD score reflects the area unit's overall level of disadvantage based on a wide range of socioeconomic factors, including education, occupation, income, unemployment, household structure, and household tenure (plus others). For analysis, the SA1s in each capital city were grouped into quartiles based on their IRSD scores, with Q1 denoting the 25% least disadvantaged areas in each city and Q4 the most disadvantaged 25%. SA1s are geographic areas that form part of the Australian Statistical Geographic Standard [41]. SA1s are the smallest unit of geography for which Census data are available, and across the whole of Australia their resident population ranges from 200–800 people, and an average of about 445 people in metropolitan cities. SA1s have built environments that can include commercial and retail development, educational facilities, local park, playgrounds, and public open space, among other infrastructure [41]

**Residential distance to the CBD.** The 2017–18 NHS included residential SA1 information but not actual residential distance from the CBD. Hence, we used SA1 boundaries from a separate data source (the 2016 Australian Population and Housing Census), to derive a measure of SA1 distances to the CBD and incorporated this into the NHS by merging on an SA1 identifier common in both datasets. Specifically, we used MapInfo RouteFinder (MapInfo Pro

v2019.1, RouteFinder v6.01, Precisely, Pearl River, New York) to calculate the Euclidean distance (km) from the centroid of each SA1 to the city's General Post Office point (co-ordinates extracted from Google Maps [Google Inc, Mountain View, California]), which in the Australian context, denotes the CBD. Residential distance to the CBD was operationalised as a continuous variable, with some neighbourhoods being very close to the CBD and others being very distant. We know from previous research that suburbs on the outskirts of Australia's capital cities are more likely to be sprawling, and to have a built form that is less residentially dense, with a poorly interconnected street network, and a less diverse land use mix [42]. These types of neighbourhoods are less walkable, they are often located a long way from employment and essential services, they require greater reliance on private motor vehicles, and are less conducive to the use of public transport and active travel (i.e., waking and cycling). In short, neighbourhoods further from the city are more likely to be obesogenic, and residents of these neighbourhoods are more likely to be overweight or obese [37, 43, 44].

**Supermarket availability.** Consistent with previous Australian research [45], this was defined as the count of major chain supermarkets within a 1000m road-network buffer centred on each participant's place of residence, and included ALDI, Coles, Foodland, Food Works, Franklins, Fresh Market, Friendly Grocer, IGA, Safeway, and Woolworths. Supermarket data were sourced from HERE data (MapData Services, Sydney, Australia, 2018). Supermarket availability reflects the opportunity to purchase a wide variety of foods, including healthier options, and is generally considered to represent healthful food availability [46].

**Availability of walkable amenities.** This was defined as the count of amenities within a 1000m road-network buffer centred on each participant's place of residence, and included libraries, petrol stations, hospitality, retail (excluding supermarkets), and entertainment venues. These amenities reflect local destinations that participants may walk to, and their proximity is consistent with the amenity structure of 20-minute neighbourhoods [47].

To maintain confidentiality of NHS participant information, geocoding, spatial association and the construction of buffers and environmental measures were conducted by the ABS. The ABS created the network buffers and environmental measures (i.e., Supermarket availability, Availability of walkable amenities) using ArcGIS Network Analyst (Esri, version 9.3.1; Redlands, CA, USA).

## Outcome measures

**Body size.** Two measures were obtained by direct physical measurement via standard procedures by trained ABS personnel: (1) body mass index (BMI) (calculated as weight [kg] divided by height $[m]^2$), and (2) waist circumference (cm). Interviewers used a digital scale to measure weight, a stadiometer to measure height, and a non-extensible, metal tape measure to measure waist circumference. Given respondent sensitivities about physically being measured, non-response rates were relatively high (33.8% for BMI, and 35.4% for waist circumference). Prior to the release of the NHS dataset, the ABS imputed missing values for these measures using the Hot Deck method [48]. In brief, this involved matching a 'recipient' (i.e., a record with missing physical data) with a 'donor' (i.e., a non-missing record with similar characteristics as the recipient) using sex, age-group, location, self-perceived body mass, level of exercise, cholesterol status as a long-term condition, and BMI collected from self-report height and weight. The ABS conducted sensitivity tests finding that differences between the 'measured only' data and the 'measured and imputed' data were negligible, with the latter deemed by the ABS to be of 'suitable quality' [39]. Therefore, consistent with previous Australian research, [44, 45] this study used 'measured and imputed' body size data as provided by the ABS.

## Covariates

Potential confounders of the associations between neighbourhood disadvantage and body size were identified according to our DAG and included: age (years); sex (male or female); country of birth, initially categorised into 10 regions using the Standard Australian Classification of Countries [49] and subsequently grouped as Australian-born and overseas-born; highest educational qualification completed, coded as bachelor's degree or higher, diploma, certificate, or high school or less; occupation, initially categorised in accordance with the ABS' Australian and New Zealand Standard Classification of Occupations [50] and subsequently coded as manager and professional, white collar employee, blue collar worker, and 'not in the labour market' (e.g. unemployed, retired, home duties); and total equivalised household income categorised into quintiles, with households in Q5 being in the lowest income category.

## Analytic dataset and analytic approach

Delimiting the NHS sample to residents of SA1s in state and territory capital cities reduced the sample from n = 21,315 to n = 13,078. Of this portion, we retained persons aged 18 years and over (n = 9,881). Further, we excluded women who reported being pregnant (n = 45), persons with missing data for count of supermarkets or amenities (n = 137) and persons with missing data on one or more confounding variables (n = 302). The final analytic sample comprised 9,367 persons spread across 3,454 SA1s.

To support our secondary aim of assessing generalisability of findings from one city to other cities, and due to findings from previous studies using the NHS data which indicated between-city variation in the relationships between environmental indicators and body size [44, 45] analyses in this study were stratified by city. A multi-level modelling approach was used with the unit of analysis at the level of the individual, adjusting for covariates and accounting for clustering of individuals within SA1's. Clustering of participants within geographical location creates issues of non-independence due to correlations between measures. Multilevel models can account for this non-independence [51]. The first level modelled participant values of body size as a function of the obesogenic environment, while a second level accounted for spatial clustering within SA1, with a random intercept specified to allow for variations in values of body size across SA1's. Analyses involved four stages for each capital city. First, the sociodemographic, built environment, and anthropometric characteristics of the analytic sample were described using univariate statistics. Second, ANOVA was used to examine bivariate associations between body size, residential distance to the CBD, and the built environment, and neighbourhood disadvantage quartiles. Based on the published literature we expected residents of disadvantaged neighbourhoods would have higher average BMI and a larger waist circumference than residents of advantaged neighbourhoods, and that disadvantaged neighbourhoods would be located further away from the CBD and have fewer supermarkets and walkable amenities. Third, bivariate linear regression models (adjusted for participant clustering within SA1 units) were used to examine associations between residential distance to the CBD, the built environment, and body size. We expected that living further away from the CBD would be positively associated with body size, and that having more walkable access to supermarkets and amenities would be negatively associated with body size. Fourth, multivariable linear regression was used to estimate associations between neighbourhood disadvantage and body size, adjusting for individual level covariates and accounting for clustering of individuals within SA1s (Model 1). This baseline analysis was then extended based on the DAG to include residential distance to the CBD (Model 2), and we expected to observe an attenuation of the association between neighbourhood disadvantage and body size. Model 2 was further extended by including count of supermarkets (Model 3), then count of walkable amenities

(replacing count of supermarkets) (Model 4), and then simultaneous adjustment for supermarkets and amenities (Model 5). We expected to see a further attenuation of the relationship between body size and neighbourhood disadvantage due to differences between advantaged and disadvantaged neighbourhoods in the availability of supermarkets and walkable amenities. The extent of attenuation was denoted (indicatively) as a percentage reduction in the magnitude of neighbourhood differences in body size for Models 2–5 compared with Model 1. Model outputs are expressed as β coefficients and their 95% confidence intervals. All data preparation and analyses were conducted using Stata version 16.0 for Windows [52].

## Results

Table 1 presents univariate descriptive statistics showing the sociodemographic, built environment, and body size characteristics of the analytic sample for each capital city. There were notable differences between cities in terms of age, sex, and country of birth, and in how education, occupation, household income, and neighbourhood disadvantage were distributed by city. Neighbourhoods in Sydney were located an average of 20km from the CBD, and in Hobart the corresponding distance was 7.5km. Supermarket availability varied widely between cities, ranging from a mean availability of 0.3 supermarkets within 1000m of respondents' residences in Hobart, to 0.92 supermarkets per 1000m in Sydney. Walkable amenities within 1000m of respondents' residences was lowest in Canberra (an average of 19.0 amenities) and highest in Sydney (111.3). Mean BMI ranged from 27.4 kg/m$^2$ in Sydney to 28.5 kg/m$^2$ in Brisbane: the mean waist circumference ranged from 91.5cm in Darwin to 94.4cm in Adelaide.

Table 2 presents bivariate associations (from ANOVAs) between body size, residential distance to the CBD, and densities of supermarkets and walkable amenities within 1000m of participants' residences, and neighbourhood disadvantage. Neighbourhood disadvantage was statistically significantly associated with body size except in Hobart (non-significant for waist circumference), Darwin and Canberra (non-significant for both BMI and waist circumference). Associations were generally monotonic rather than linear and showed that BMI and waist circumference were typically lowest in the least disadvantaged neighbourhoods and highest in the most disadvantaged neighbourhoods (Hobart, Darwin and Canberra tended to this same pattern).

For each capital city except Darwin, there were statistically significant associations between neighbourhood disadvantage and residential distance to the CBD: socioeconomically disadvantaged neighbourhoods were more likely to be located further away from a city's CBD.

In all capital cities except Brisbane, there was a statistically significant association between neighbourhood disadvantage and supermarket density; however, the association showed a mixed and complex pattern. In Melbourne, Perth, and Canberra, the most disadvantaged neighbourhoods had the largest average number of supermarkets within 1000m of participants' residences. By contrast, the greatest supermarket availability was in Q2 for Sydney, Adelaide, and Hobart; Q2 and Q3 for Darwin.

Neighbourhood disadvantage was also statistically significantly associated with walkable amenities in all capital cities except Perth; again however, the patterns were complex and mixed. In Melbourne, Brisbane, and Adelaide, count of walkable amenities was lowest in the most disadvantaged neighbourhoods. By contrast, in Sydney, Hobart, and Darwin, the lowest count of walkable amenities was found in the least disadvantaged neighbourhoods. For Canberra both Q1 and Q3 had similarly low walkable amenities counts.

Table 3 presents bivariate associations (multilevel linear regression) between residential distance to the CBD, supermarkets count, count of walkable amenities, and body size, by capital city. Both BMI and waist circumference were statistically significantly associated with residential distance to the CBD, for BMI this was all capital cities except Darwin, for waist

**Table 1. Sociodemographic, built environment, and body size characteristics of the analytic sample by capital city.**

| | Sydney | Melbourne | Brisbane | Adelaide | Perth | Hobart | Darwin | Canberra |
|---|---|---|---|---|---|---|---|---|
| Number of individuals | 1,675 | 1,735 | 1,383 | 1,067 | 1,173 | 533 | 718 | 1,050 |
| Number of areas (SA1) | 719 | 614 | 550 | 487 | 437 | 171 | 126 | 350 |
| *Age* (mean, SD) | 49.8 (18.0) | 48.8 (18.5) | 48.1 (17.9) | 52.0 (18.3) | 49.6 (18.1) | 52.0 (18.1) | 45.1 (16.0) | 48.1 (17.6) |
| *Sex* (% men) | 45.4 | 46.9 | 45.4 | 46.0 | 46.3 | 40.1 | 49.5 | 43.0 |
| *Country of birth* | | | | | | | | |
| Australian born (%) | 49.1 | 57.6 | 68.6 | 68.8 | 54.6 | 80.3 | 63.5 | 66.7 |
| *Education (%)* | | | | | | | | |
| Bachelor's degree+ | 38.7 | 38.1 | 29.9 | 27.2 | 29.7 | 32.4 | 31.2 | 46.1 |
| Diploma | 11.1 | 12.0 | 12.8 | 10.5 | 12.6 | 10.6 | 10.3 | 11.4 |
| Certificate | 15.4 | 14.5 | 22.0 | 19.5 | 19.8 | 21.9 | 24.3 | 12.7 |
| High school or less | 34.6 | 35.2 | 35.1 | 42.5 | 37.7 | 34.9 | 34.1 | 29.7 |
| *Occupation (%)* | | | | | | | | |
| Managers and professionals | 27.8 | 28.8 | 24.0 | 24.7 | 24.3 | 23.4 | 23.8 | 37.1 |
| White collar | 19.1 | 20.1 | 21.6 | 18.5 | 17.7 | 20.2 | 27.7 | 21.5 |
| Blue collar | 14.1 | 14.5 | 17.7 | 14.4 | 19.1 | 13.8 | 21.5 | 10.4 |
| Not in labor market | 38.8 | 36.4 | 36.5 | 42.2 | 38.7 | 42.4 | 26.8 | 30.8 |
| *Household income (quintiles) (%)* | | | | | | | | |
| Q1 (highest) | 15.8 | 13.8 | 17.1 | 22.6 | 18.3 | 22.5 | 14.3 | 11.1 |
| Q2 | 13.3 | 16.4 | 16.1 | 19.7 | 13.9 | 20.5 | 12.2 | 12.1 |
| Q3 | 15.1 | 16.4 | 18.4 | 17.8 | 17.3 | 18.2 | 17.4 | 15.6 |
| Q4 | 16.7 | 17.9 | 16.2 | 16.7 | 17.3 | 16.8 | 19.0 | 20.1 |
| Q5 (lowest) | 22.9 | 15.9 | 17.2 | 14.5 | 16.7 | 14.0 | 25.5 | 29.2 |
| Missing | 15.9 | 19.3 | 14.8 | 8.4 | 16.1 | 8.2 | 11.4 | 11.6 |
| *Neighbourhood disadvantage[1] (quartiles) (%)* | | | | | | | | |
| Q1 (least disadvantaged) | 39.8 | 28.6 | 25.6 | 27.9 | 25.5 | 28.1 | 24.2 | 20.8 |
| Q2 | 31.3 | 40.4 | 35.2 | 33.9 | 33.0 | 30.9 | 25.0 | 33.9 |
| Q3 | 17.6 | 18.5 | 23.5 | 23.6 | 25.6 | 23.2 | 32.7 | 29.9 |
| Q4 (most disadvantaged) | 11.1 | 12.3 | 15.4 | 14.5 | 15.6 | 17.6 | 17.9 | 15.3 |
| Residential distance to the CBD[2] (km) | 20.8 (13.2) | 20.5 (13.5) | 18.0 (10.4) | 12.0 (7.5) | 18.8 (13.9) | 7.5 (4.6) | 10.6 (5.0) | 10.2 (4.2) |
| Supermarkets[3] (No. per 1000m buffer) | 0.92 (1.28) | 0.85 (1.35) | 0.63 (0.98) | 0.77 (1.07) | 0.55 (0.84) | 0.30 (0.57) | 0.43 (0.90) | 0.38 (0.68) |
| Walkable amenities[3] (No. per 1000m buffer) | 111.3 (249.2) | 85.5 (229.8) | 37.7 (81.1) | 46.3 (75.0) | 25.3 (44.9) | 38.1 (71.5) | 26.9 (47.6) | 19.0 (53.4) |
| Body mass index, kg/m$^2$ (mean, SD) | 27.4 (5.7) | 27.7 (5.5) | 28.5 (6.1) | 28.4 (5.8) | 27.9 (5.7) | 27.7 (5.7) | 27.6 (5.8) | 27.8 (5.9) |
| Waist circumference, cm (mean, SD) | 91.7 (15.5) | 92.0 (14.8) | 94.2 (16.2) | 94.4 (15.8) | 92.4 (14.9) | 91.6 (14.9) | 91.5 (16.6) | 92.7 (15.7) |

[1]. Areas (SA1s) were grouped into city-specific quartiles (25%) based on their Index of Relative Socioeconomic Disadvantage score. The number in the table denotes the percentage of participants who resided in areas classified in each quartile

[2]. Measured as the Euclidean distance (km) from the centroid of each participant's area of residence (SA1) to the capital city central business district (CBD).

[3]. Number of supermarkets and walkable amenities within a 1000m road network buffer from each participant's residence

circumference this was all except Hobart, Darwin, and Canberra. Residents of neighbourhoods that were further from the CBD had higher BMI and larger waist circumference than residents of neighbourhoods located closer to the CBD.

There was a statistically significant negative association between supermarket density and body size in Sydney, Melbourne, Brisbane, and Hobart: a larger number of supermarkets within 1000m of participants' residences was associated with lower BMI and smaller waist circumference. There were no associations between supermarket density and body size in Adelaide, Perth, Darwin, and Canberra.

**Table 2. Body mass index (kg/m², waist circumference (cm), residential distance to the CBD, supermarket count, and walkable amenities, by neighbourhood disadvantage and capital city (bivariate associations using ANOVAs).**

| | Body mass index (kg/m²) | Waist circumference (cm) | Residential distance to the CBD (km)[1] | Supermarkets (No. per 1000m)² | Walkable amenities (No. per 1000m)² |
|---|---|---|---|---|---|
| *Sydney (individuals n = 1,675, areas n = 719)* | | | | | |
| Q1 (least disadvantaged) | 26.9 (26.5, 27.2) | 90.4 (89.3, 91.5) | 17.0 (15.5, 18.5) | 0.70 (0.62, 0.79) | 77.3 (65.7, 88.9) |
| Q2 | 27.6 (27.1, 28.1) | 92.0 (90.6, 93.4) | 20.1 (18.4, 21.8) | 1.11 (0.99, 1.23) | 128.1 (106.0, 150.2) |
| Q3 | 27.7 (27.0, 28.4) | 93.0 (91.2, 94.9) | 24.1 (21.9, 26.4) | 0.95 (0.81, 1.10) | 142.8 (100.4, 185.2) |
| Q4 (most disadvantaged) | 28.2 (27.2, 29.2) | 93.4 (90.9, 95.9) | 23.0 (20.7, 25.2) | 1.10 (0.89, 1.30) | 136.0 (100.2, 171.9) |
| p-value | 0.011 | 0.025 | <0.0001 | <0.0001 | <0.0001 |
| *Melbourne (individuals n = 1,735, areas n = 614)* | | | | | |
| Q1 (least disadvantaged) | 27.2 (26.7, 27.7) | 90.6 (89.3, 91.9) | 15.2 (13.6, 16.9) | 0.88 (0.77, 0.99) | 113.4 (100.5, 126.7) |
| Q2 | 27.5 (27.1, 27.9) | 91.8 (90.7, 92.9) | 21.4 (19.6, 23.2) | 0.83 (0.72, 0.93) | 74.0 (55.0, 93.1) |
| Q3 | 28.2 (27.6, 28.9( | 93.7 (92.1, 95.4) | 23.7 (21.2, 26.2) | 0.70 (0.55, 0.86) | 79.0 (46.0, 111.9) |
| Q4 (most disadvantaged) | 28.2 (27.4, 29.0) | 93.5 (91.4, 95.6) | 21.0 (18.3, 23.8) | 1.06 (0.87, 1.25) | 68.0 (49.6, 86.4) |
| p-value | 0.030 | 0.012 | <0.0001 | 0.025 | 0.014 |
| *Brisbane (individuals n = 1,383, areas n = 550)* | | | | | |
| Q1 (least disadvantaged) | 28.1 (27.5, 28.7) | 92.8 (91.1, 94.5) | 11.9 (10.6, 13.2) | 0.63 (0.51, 0.75) | 31.9 (23.8, 40.0) |
| Q2 | 28.0 (27.5, 28.5) | 92.9 (91.6, 94.3) | 15.1 (13.8, 16.4) | 0.59 (0.52, 0.67) | 37.9 (31.0, 44.7) |
| Q3 | 28.5 (27.9, 29.2) | 95.2 (93.4, 96.9) | 21.8 (20.1, 23.6) | 0.72 (0.60, 0.83) | 51.2 (39.9, 62.6) |
| Q4 (most disadvantaged) | 30.4 (29.4, 31.4) | 98.0 (95.5, 100.5) | 25.6 (23.6, 27.6) | 0.54 (0.42, 0.66) | 26.5 (20.4, 32.5) |
| p-value | <0.0001 | <0.001 | <0.0001 | 0.170 | 0.002 |
| *Adelaide (individuals n = 1,067, areas n = 487)* | | | | | |
| Q1 (least disadvantaged) | 27.9 (27.3, 28.5) | 92.5 (90.8, 94.2) | 9.8 (8.9, 10.7) | 0.59 (0.48, 0.70) | 43.3 (34.7, 51.8) |
| Q2 | 27.9 (27.4, 28.5) | 93.8 (92.3, 95.3) | 11.0 (9.8, 12.1) | 0.90 (0.79, 1.00) | 56.1 (47.2, 65.1) |
| Q3 | 29.1 (28.4, 29.9) | 96.9 (94.8, 99.0) | 13.9 (12.2, 15.6) | 0.84 (0.68, 1.00) | 41.9 (34.3, 49.5) |
| Q4 (most disadvantaged) | 29.3 (28.1, 30.5) | 95.6 (92.9, 98.3) | 17.0 (15.1, 18.9) | 0.67 (0.54, 0.81) | 36.6 (26.9, 46.4) |
| p-value | 0.006 | 0.007 | <0.0001 | 0.001 | 0.017 |
| *Perth (individuals n = 1,173, areas n = 437)* | | | | | |
| Q1 (least disadvantaged) | 26.9 (26.4, 27.5) | 90.6 (89.1, 92.2) | 14.4 (12.3, 16.5) | 0.52 (0.44, 0.60) | 27.5 (23.7, 31.3) |
| Q2 | 27.9 (27.3, 28.4) | 92.1 (90.6, 93.6) | 17.3 (15.0, 19.5) | 0.39 (0.32, 0.47) | 23.6 (18.0, 29.1) |
| Q3 | 28.1 (27.4, 28.7) | 92.8 (91.1, 94.5) | 20.1 (16.8, 23.4) | 0.62 (0.51, 0.73) | 22.6 (18.8, 26.3) |
| Q4 (most disadvantaged) | 29.1 (28.2, 30.0) | 95.1 (92.9, 97.4) | 19.7 (16.6, 22.8) | 0.80 (0.65, 0.94) | 29.6 (22.2, 37.1) |
| p-value | <0.001 | 0.014 | 0.014 | <0.0001 | 0.250 |
| *Hobart (individuals n = 533, areas n = 171)* | | | | | |
| Q1 (least disadvantaged) | 27.5 (26.6, 28.5) | 91.8 (89.1, 94.5) | 6.2 (4.9, 7.4) | 0.10 (0.05, 0.15) | 21.8 (15.4, 28.1) |
| Q2 | 26.8 (26.1, 27.6) | 89.8 (87.7, 91.9) | 7.2 (5.8, 8.6) | 0.49 (0.39, 0.60) | 43.4 (32.5, 54.2) |
| Q3 | 27.9 (27.0, 28.8) | 92.0 (89.7, 94.3) | 7.3 (5.8, 8.9) | 0.33 (0.22, 0.45) | 51.2 (35.3, 67.1) |
| Q4 (most disadvantaged) | 29.4 (28.0, 30.8) | 93.9 (90.7, 97.1) | 9.2 (7.7, 10.8) | 0.21 (0.11, 0.31) | 37.8 (21.5, 54.0) |
| p-value | 0.006 | 0.192 | 0.042 | <0.0001 | 0.005 |
| *Darwin (individuals n = 718, areas n = 126)* | | | | | |
| Q1 (least disadvantaged) | 27.0 (26.2, 27.9) | 89.4 (86.8, 91.9) | 9.8 (7.6, 11.9) | 0.12 (0.03, 0.20) | 13.7 (9.3, 18.0) |
| Q2 | 27.2 (26.5, 28.0) | 91.1 (88.7, 93.5) | 9.2 (7.2, 11.2) | 0.54 (0.38, 0.70) | 34.4 (26.1, 42.7) |
| Q3 | 27.8 (27.1, 28.6) | 92.8 (90.8, 94.9) | 10.9 (9.4, 12.4) | 0.54 (0.42, 0.66) | 31.6 (24.7, 38.5) |

*(Continued)*

**Table 2.** (Continued)

| | Body mass index (kg/m²) | Waist circumference (cm) | Residential distance to the CBD (km)[1] | Supermarkets (No. per 1000m)[2] | Walkable amenities (No. per 1000m)[2] |
|---|---|---|---|---|---|
| Q4 (most disadvantaged) | 28.2 (27.0, 29.4) | 92.3 (89.5, 95.1) | 12.1 (10.6, 13.7) | 0.48 (0.34, 0.61) | 25.5 (19.0, 32.0) |
| p-value | 0.232 | 0.184 | 0.176 | <0.0001 | <0.0001 |
| *Canberra (individuals n = 1,050, areas n = 350)* | | | | | |
| Q1 (least disadvantaged) | 27.6 (26.7, 28.4) | 93.0 (89.1, 92.2) | 8.8 (7.8, 9.8) | 0.22 (0.16, 0.29) | 13.1 (10.3, 16.0) |
| Q2 | 27.8 (27.2, 28.5) | 92.2 (90.6, 93.6) | 9.8 (9.1, 10.6) | 0.40 (0.32, 0.47) | 21.2 (15.6, 26.9) |
| Q3 | 27.9 (27.3, 28.5) | 92.8 (91.1, 94.5) | 10.9 (10.2, 11.7) | 0.42 (0.34, 0.50) | 13.0 (10.2, 15.9) |
| Q4 (most disadvantaged) | 28.0 (27.1, 29.0) | 93.2 (92.9, 97.4) | 10.0 (8.8, 11.1) | 0.44 (0.33, 0.55) | 33.5 (18.0, 48.9) |
| p-value | 0.864 | 0.911 | 0.012 | 0.003 | 0.0003 |

[1]. Measured as the Euclidian distance (km) from the centroid of each participant's area of residence (SA1) to the capital city central business district (CBD).

[2]. Number of supermarkets and walkable amenities within a 1000m road network buffer from each participant's residence

Lastly, walkable amenities and body size were generally negatively associated: BMI was lower and waist circumference smaller in areas with a larger number of local amenities in Sydney, Brisbane, Perth (BMI only), and Canberra. Unexpectedly, waist circumference was larger with more local amenities in Darwin.

Table 4 presents associations between neighbourhood disadvantage and BMI before and after adjustment for residential distance to the CBD, count of supermarkets, and walkable amenities. Residents of socioeconomically disadvantaged neighbourhoods in Sydney, Melbourne, Brisbane, Adelaide, and Perth had significantly higher BMI than residents of the least disadvantaged neighbourhoods (Model 1). Associations between neighbourhood disadvantage and BMI in these capital cities were attenuated after adjustment for neighbourhood proximity to the city's CBD (Model 2). The magnitude of difference in BMI between the least and most disadvantaged neighbourhoods in Brisbane reduced from 2.11 kg/m² to 1.51 kg/m² (28.5%) after adjustment for residential distance to the CBD. The corresponding reductions in the magnitude of neighbourhood inequalities in BMI in Adelaide, Melbourne, Sydney, and Perth were 15.4%, 10.5%, 9.0%, and 4.0% respectively. Though coefficients changed relative to

**Table 3. Associations[1] between residential distance to the CBD, the food environment, walkable amenities, and body mass index and waist circumference, by capital city (bivariate, multilevel linear regression).**

| | Body Mass Index (kg/m²) | | | Waist circumference (cm) | | |
|---|---|---|---|---|---|---|
| | Residential location (km)[2] | Supermarkets (No. per 1000m)[3] | Walkable amenities (No. per 1000m)[3] | Residential location (km)[2] | Supermarkets (No. per 1000m)[3] | Walkable amenities (No. per 1000m)[3] |
| Sydney | 0.060 (0.03, 0.08) | -0.319 (-0.52, -0.11) | -0.002 (-0.003, -0.001) | 0.130 (0.06, 0.19) | -0.653 (-1.20, -0.09) | -0.005 (-0.007, -0.002) |
| Melbourne | 0.058 (0.03, 0.07) | -0.322 (-0.53, -0.10) | -0.001 (-0.003, .009) | 0.152 (0.10, 0.20) | -0.847 (-1.56, -0.13) | -0.003 (-0.009, 0.002) |
| Brisbane | 0.076 (0.04, 0.10) | -0.586 (-0.89, -0.27) | -0.006 (-0.010, -0.001) | 0.145 (0.06, 0.26) | -1.27 (-2.05, -0.50) | -0.014 (-0.02, -003) |
| Adelaide | 0.088 (0.04, 0.13) | -0.138 (-0.41, 0.13) | -0.003 (-0.007, 0.004) | 0.282 (0.15, 0.41) | -0.079 (-0.87, 0.71) | -0.010 (-0.02, 0.001) |
| Perth | 0.031 (0.006, 0.056) | 0.050 (-0.31, 0.041) | -0.005 (-0.011, -0.001) | 0.059 (0.002, 0.11) | -0.110 (-1.04, 0.82) | -0.019 (-0.03, 0.003) |
| Hobart | 0.122 (0.02, 0.22) | -1.10 (-1.73, -0.047) | -0.003 (-0.01, 0.008) | 0.187 (-0.05, 0.43) | -3.36 (-5.00, -1.71) | -0.014 (-0.036, 0.008) |
| Darwin | 0.015 (-0.07, 0.10) | 0.130 (-0.32, 0.58) | 0.007 (-0.001, 0.017) | 0.013 (-0.27, 0.30) | 0.961 (-0.25, 2.17) | 0.030 (0.003, 0.058) |
| Canberra | 0.116 (0.03, 0.19) | -0.332 (-0.92, 0.25) | -0.008 (-0.013, -0.002) | 0.200 (-0.03, 0.44) | -1.39 (-2.89, 0.11) | -0.017 (-0.03, -0.002) |

[1]. Multilevel linear regression coefficients and 95% confidence intervals. Modelling adjusted for participant clustering within SA1 units.

[2]. Euclidian distance (km) from the centroid of each participant's area of residence (SA1) to the capital city central business district (CBD).

[3]. Number of supermarkets and walkable amenities within a 1000m road network buffer from each participant's residence

**Table 4. Neighbourhood disadvantage and body mass index (kg/m$^2$), before and after adjustment for residential distance to the CBD, and count of supermarkets and walkable amenities, by capital city (multilevel linear regression).**

| | Model 1[a, b] | Model 2[c] | Model 3[d] | Model 4[e] | Model 5[f] |
|---|---|---|---|---|---|
| *Sydney (individuals n = 1,675, areas n = 719)* | | | | | |
| Q1 (least disadvantaged) | - - | - - | - - | - - | - - |
| Q2 | 0.84 (0.21, 1.49) | 0.80 (0.16, 1.43) | 0.85 (0.21, 1.49) | 0.86 (0.22, 1.49) | 0.86 (0.22, 1.50) |
| Q3 | 1.35 (0.51, 2.19) | 1.07 (0.25, 1.89) | 1.12 (0.29, 1.94) | 1.17 (0.34, 2.00) | 1.17 (0.34, 2.00) |
| Q4 (most disadvantaged) | 1.79 (0.64, 2.94) | 1.63 (0.49, 2.77) | 1.68 (0.52, 2.84) | 1.71 (0.56, 2.85) | 1.71 (0.56, 2.87) |
| *Melbourne (individuals n = 1,735, areas n = 614)* | | | | | |
| Q1 (least disadvantaged) | - - | - - | - - | - - | - - |
| Q2 | 0.42 (-0.19, 1.05) | 0.18 (-0.46, 0.82) | 0.20 (-0.44, 0.84) | 0.18 (-0.46, 0.82) | 0.23 (-0.40, 0.88) |
| Q3 | 1.10 (0.29, 1.91) | 0.80 (-0.04, 1.66) | 0.82 (-0.20, 1.67) | 0.80 (-0.04, 1.65) | 0.83 (-0.01, 1.68) |
| Q4 (most disadvantaged) | 1.24 (0.27, 2.21) | 1.11 (0.15, 2.07) | 1.16 (0.20, 2.11) | 1.11 (0.15, 2.07) | 1.25 (0.28, 2.22) |
| *Brisbane (individuals n = 1,383, areas n = 550)* | | | | | |
| Q1 (least disadvantaged) | - - | - - | - - | - - | - - |
| Q2 | -0.06 (-0.78, 0.65) | -0.19 (-0.93, 0.54) | -0.16 (-0.88, 0.56) | -0.14 (-0.87, 0.58) | -0.16 (-0.88, 0.55) |
| Q3 | 0.20 (-0.74, 1.16) | -0.12 (-1.08, 0.83) | 0.04 (-0.90, 1.00) | 0.01 (-0.97, 1.01) | 0.03 (-0.94, 1.01) |
| Q4 (most disadvantaged) | 2.11 (1.04, 3.18) | 1.51 (0.31, 2.72) | 1.70 (0.50, 2.90) | 1.65 (0.42, 2.88) | 1.69 (0.48, 2.91) |
| *Adelaide (individuals n = 1,067, areas n = 487)* | | | | | |
| Q1 (least disadvantaged) | - - | - - | - - | - - | - - |
| Q2 | -0.01 (-0.75, 0.72) | -0.03 (-077, 0.70) | -0.04 (-0.79, 0.70) | -0.04 (-0.78, 0.69) | -0.04 (-0.79, 0.70) |
| Q3 | 1.12 (0.20, 2.05) | 1.00 (0.07, 1.92) | 0.99 (0.05, 1.93) | 0.99 (0.07, 1.92) | 0.99 (0.05, 1.93) |
| Q4 (most disadvantaged) | 1.37 (0.15, 2.59) | 1.16 (-0.05, 2.37) | 1.15 (-0.06, 2.37) | 1.15 (-0.06, 2.37) | 1.15 (-0.06, 2.37) |
| *Perth (individuals n = 1,173, areas n = 437)* | | | | | |
| Q1 (least disadvantaged) | - - | - - | - - | - - | - - |
| Q2 | 1.19 (0.40, 1.98) | 1.11 (0.81, 3.08) | 1.11 (0.33, 1.90) | 1.11 (0.33, 1.89) | 1.12 (0.34, 1.90) |
| Q3 | 1.08 (0.24, 1.92) | 0.95 (0.10, 1.81) | 0.93 (0.07, 1.79) | 0.96 (0.11, 1.81) | 0.92 (0.07, 1.78) |
| Q4 (most disadvantaged) | 2.03 (0.88, 3.18) | 1.95 (0.81, 3.08) | 1.92 (0.78, 3.05) | 1.97 (0.84, 3.11) | 1.91 (0.77, 3.05) |
| *Hobart (individuals n = 533, areas n = 171)* | | | | | |
| Q1 (least disadvantaged) | - - | - - | - - | - - | - - |
| Q2 | -0.66 (-1.77, 0.43) | -0.78 (-1.90, 0.33) | -0.49 (-1.71, 0.73) | -0.83 (-2.05, 0.38) | -0.55 (-1.82, 0.70) |
| Q3 | 0.19 (-1.03, 1.42) | 0.11 (-1.06, 1.29) | 0.32 (-0.85, 1.51) | 0.05 (-1.23, 1.33) | 0.20 (-1.04, 1.46) |
| Q4 (most disadvantaged) | 1.32 (-0.29, 2.95) | 1.19 (-0.44, 2.83) | 1.37 (-0.30, 3.04) | 1.13 (-0.50, 2.76) | 1.24 (-0.40, 2.90) |
| *Darwin (individuals n = 718, areas n = 126)* | | | | | |
| Q1 (least disadvantaged) | - - | - - | - - | - - | - - |
| Q2 | 0.004 (-1.27, 1.28) | -0.001 (-1.28, 1.28) | -0.06 (-1.31, 1.18) | -0.17 (-1.44, 1.08) | -0.12 (-1.36, 1.12) |
| Q3 | 0.54 (-0.53, 1.62) | 0.52 (-0.57, 1.61) | 0.43 (-0.68, 1.56) | 0.30 (-0.78, 1.40) | 0.37 (-0.73, 1.47) |
| Q4 (most disadvantaged) | 1.07 (-0.26, 2.41) | 1.05 (-0.28, 2.39) | 0.98 (-0.35, 2.32) | 0.89 (-0.44, 2.23) | 0.96 (-0.39, 2.33) |
| *Canberra (individuals n = 1,050, areas n = 350)* | | | | | |
| Q1 (least disadvantaged) | - - | - - | - - | - - | - - |
| Q2 | 0.39 (-0.69, 1.48) | 0.28 (-0.79, 1.37) | 0.33 (-0.75, 1.43) | 0.32 (-0.76, 1.41) | 0.34 (-0.74, 1.44) |
| Q3 | 0.42 (-0.66, 1.51) | 0.28 (-0.78, 1.35) | 0.33 (-0.70, 1.38) | 0.29 (-0.76, 1.36) | 0.32 (-0.71, 1.37) |
| Q4 (most disadvantaged) | 0.41 (-0.83, 1.66) | 0.35 (-0.86, 1.57) | 0.40 (-0.81, 1.61) | 0.42 (-0.76, 1.41) | 0.43 (-0.77, 1.64) |

[a]. Multilevel linear regression coefficients and 95% confidence intervals. Modelling adjusted for participant clustering within SA1 units.

[b]. Model 1: Neighbourhood disadvantage and body mass index adjusting for age, sex, country of birth, education, occupation, and household income

[c]. Model 2: Model 1 plus adjustment for residential distance to the CBD (measured as the Euclidean distance between the centroid of each participant's area of residence and the capital city Central Business District)

[d]. Model 3: Model 2 plus adjustment for count of supermarkets within a 1000m buffer of the participants place of residence

[e]. Model 4: Model 2 plus adjustment for count of walkable amenities within a 1000m buffer of the participants place of residence

[f]. Model 5: Model 2 plus adjustment for count of supermarkets and walkable amenities within a 1000m buffer of the participants place of residence

Model 2 (some remained similar, some increased) on inclusion of either count of supermarkets (Model 3) or walkable amenities (Model 4) associations between neighbourhood disadvantage and BMI in Sydney, Melbourne, Brisbane, Adelaide, and Perth remained attenuated relative to baseline (Model 1).The reductions in inequalities between the least and most disadvantaged neighbourhoods ranged from 5.5% in Perth to 19.5% in Brisbane with adjustment for residential distance to the CBD and supermarkets, and from 3.0% in Sydney to 21.9% in Brisbane with adjustment for location and amenities. After adjustment for residential distance to the CBD, as well as both supermarkets and walkable amenities (Model 5), associations between neighbourhood disadvantage and BMI were attenuated relative to baseline (Model 1) in Sydney, Brisbane, Adelaide, and Perth, with reductions ranging in magnitude from 4.5% in Sydney to 20.0% in Brisbane. There were no statistically significant associations between neighbourhood disadvantage and BMI in Hobart, Darwin, and Canberra.

Table 5 presents associations between neighbourhood disadvantage and waist circumference before and after adjustment for residential distance to the CBD and count of supermarkets and walkable amenities. Residents of socioeconomically disadvantaged neighbourhoods in Sydney, Melbourne, Brisbane, Adelaide, and Perth had significantly higher waist circumference than residents of the least disadvantaged neighbourhoods (Model 1). Associations between neighbourhood disadvantage and waist circumference in these capital cities were attenuated after adjustment for each neighbourhood's proximity to the city's CBD (Model 2). The magnitude of difference in waist circumference between the least and most disadvantaged neighbourhoods in Adelaide attenuated (to non-significance) from 3.21cm to 2.35cm (26.8%). The corresponding reductions in the magnitude of neighbourhood inequalities in waist circumference in Brisbane, Sydney, Melbourne, and Perth were 13.6%, 9.0%, 6.5%, and 4.7% respectively.

Similar to the BMI models, coefficients changed relative to Model 2 (some remained very similar, some increased), on inclusion of either count of supermarkets (Model 3) or walkable amenities (Model 4) associations between neighbourhood disadvantage and waist circumference in Sydney, Melbourne, Brisbane, Adelaide, and Perth remained attenuated relative to baseline (Model 1). The reductions in inequalities between the least and most disadvantaged neighbourhoods ranged from 2.8% in Perth to 29.6% in Adelaide with adjustment for location and supermarkets, and from 2.2% in Perth and 27.2% in Adelaide with adjustment for location and amenities. After simultaneous adjustment for residential distance to the CBD, supermarkets, and walkable amenities (Model 5), associations between neighbourhood disadvantage and waist circumference were attenuated relative to baseline (Model 1) in Sydney, Melbourne, Brisbane, Adelaide, and Perth, with reductions ranging in magnitude from 0.9% in Melbourne to 29.3% in Adelaide. There were no statistically significant associations between neighbourhood disadvantage and waist circumference in Hobart, Darwin, and Canberra.

## Discussion

This study examined associations between neighbourhood socioeconomic disadvantage and body size in Australia's capital cities and the influence of obesogenic environments on these associations. After adjustment for key individual-level factors, in five of eight capital cities (i.e., Sydney, Melbourne, Brisbane, Adelaide, and Perth), residents of disadvantaged neighbourhoods had greater BMI and larger waist circumference than their counterparts residing in more advantaged areas, although the magnitude of the inequities varied, sometimes markedly. These results are broadly consistent with those of other Australian [8, 9] and overseas studies [10–13].

**Table 5. Neighbourhood disadvantage and waist circumference (cm), before and after adjustment for residential distance to the CBD, supermarket availability, and walkable amenities, by capital city (multilevel linear regression).**

| | Model 1[a] | Model 2[b] | Model 3[c] | Model 4[d] | Model 5[e] |
|---|---|---|---|---|---|
| *Sydney (individuals n = 1,675, areas n = 719)* | | | | | |
| Q1 (least disadvantaged) | - - | - - | - - | - - | - - |
| Q2 | 1.94 (0.28, 3.61) | 1.83 (0.19, 3.47) | 1.87 (0.20, 3.53) | 1.92 (0.26, 3.57) | 1.89 (0.22, 3.56) |
| Q3 | 3.63 (1.61, 5.64) | 3.00 (1.02, 4.99) | 3.03 (1.03, 5.03) | 3.14 (1.12, 5.16) | 3.14 (1.12, 5.15) |
| Q4 (most disadvantaged) | 3.79 (1.20, 6.39) | 3.45 (0.84, 6.05) | 3.48 (0.85, 6.12) | 3.56 (0.94, 6.18) | 3.54 (0.91, 6.17) |
| *Melbourne (individuals n = 1,735, areas n = 614)* | | | | | |
| Q1 (least disadvantaged) | - - | - - | - - | - - | - - |
| Q2 | 2.16 (0.55, 3.76) | 1.60 (-0.02, 3.24) | 1.66 (0.03, 3.29) | 1.60 (-0.03, 3.24) | 1.71 (0.07, 3.36) |
| Q3 | 3.84 (1.72, 5.95) | 3.16 (0.96, 5.37) | 3.21 (1.02, 5.40) | 3.16 (0.97, 5.36) | 3.23 (1.03, 5.42) |
| Q4 (most disadvantaged) | 4.66 (2.20, 7.13) | 4.36 (1.90, 6.81) | 4.48 (2.04, 6.93) | 4.36 (1.89, 6.82) | 4.62 (2.14, 7.10) |
| *Brisbane (individuals n = 1,383, areas n = 550)* | | | | | |
| Q1 (least disadvantaged) | - - | - - | - - | - - | - - |
| Q2 | 0.33 (-1.51, 2.19) | 0.19 (-1.69, 2.07) | 0.28 (-1.56, 2.14) | 0.34 (-1.52, 2.22) | 0.29 (-1.55, 2.14) |
| Q3 | 1.95 (-0.41, 4.31) | 1.56 (-0.87, 4.00) | 2.05 (-0.33, 4.43) | 2.01 (-0.44, 4.47) | 2.06 (-0.34, 4.47) |
| Q4 (most disadvantaged) | 5.11 (2.22, 7.99) | 4.42 (1.23, 7.62) | 4.94 (1.77, 8.12) | 4.83 (1.60, 8.06) | 4.95 (1.77, 8.14) |
| *Adelaide (individuals n = 1,067, areas n = 487)* | | | | | |
| Q1 (least disadvantaged) | - - | - - | - - | - - | - - |
| Q2 | 1.14 (-0.94, 3.23) | 1.07 (-0.99, 3.13) | 0.95 (-1.15, 3.07) | 1.06 (-1.02, 3.14) | 0.97 (-1.15, 3.09) |
| Q3 | 3.86 (1.35, 6.37) | 3.36 (0.84, 5.88) | 3.24 (0.66, 5.82) | 3.35 (0.83, 5.88) | 3.24 (0.66, 5.83) |
| Q4 (most disadvantaged) | 3.21 (0.33, 6.09) | 2.35 (-0.55, 5.27) | 2.26 (-0.67, 5.20) | 2.34 (-0.57, 5.27) | 2.27 (-0.66, 5.21) |
| *Perth (individuals n = 1,173, areas n = 437)* | | | | | |
| Q1 (least disadvantaged) | - - | - - | - - | - - | - - |
| Q2 | 2.93 (0.80, 5.05) | 2.76 (0.64, 4.88) | 2.76 (0.64, 4.89) | 2.82 (0.71, 4.93) | 2.82 (0.71, 4.93) |
| Q3 | 2.52 (0.50, 4.53) | 2.18 (0.12, 4.24) | 2.23 (0.17, 4.28) | 2.22 (0.18, 4.27) | 2.20 (0.15, 4.26) |
| Q4 (most disadvantaged) | 5.14 (2.47, 7.82) | 4.90 (2.27, 7.56) | 5.00 (2.33, 7.66) | 5.03 (2.38, 7.67) | 5.00 (2.33, 7.67) |
| *Hobart (individuals n = 533, areas n = 171)* | | | | | |
| Q1 (least disadvantaged) | - - | - - | - - | - - | - - |
| Q2 | -1.81 (-4.66, 1.03) | -2.05 (-4.95, 0.83) | -1.12 (-4.35, 1.89) | -1.94 (-5.00, 1.12) | -1.28 (-4.46, 1.89) |
| Q3 | -0.13 (-3.20, 2.92) | -0.31 (-3.30, 2.68) | 0.28 (-2.65, 3.22) | -0.17 (-3.32, 2.97) | 0.19 (-2.84, 3.23) |
| Q4 (most disadvantaged) | 0.81 (-3.19, 4.83) | 0.53 (-3.51, 4.58) | 1.04 (-3.10, 5.18) | 0.66 (-3.42, 4.76) | 0.94 (-3.19, 5.08) |
| *Darwin (individuals n = 718, areas n = 126)* | | | | | |
| Q1 (least disadvantaged) | - - | - - | - - | - - | - - |
| Q2 | 0.60 (-3.15, 4.36) | 0.56 (-3.14, 4.27) | 0.29 (-3.39, 3.99) | 0.01 (-3.66, 3.69) | 0.16 (-3.54, 3.87) |
| Q3 | 1.76 (-1.50, 5.02) | 1.51 (-1.75, 4.78) | 1.20 (-2.09, 4.50) | 0.85 (-2.40, 4.11) | 1.01 (-2.26, 4.29) |
| Q4 (most disadvantaged) | 2.34 (-1.59, 6.29) | 2.10 (-1.77, 5.98) | 1.83 (-2.05, 5.71) | 1.59 (-2.29, 5.47) | 1.78 (-2,17, 5.74) |
| *Canberra (individuals n = 1,050, areas n = 350)* | | | | | |
| Q1 (least disadvantaged) | - - | - - | - - | - - | - - |
| Q2 | 0.06 (-2.63, 2.77) | -0.15 (-2.86, 2.54) | 0.04 (-2.65, 2.74) | -0.11 (-2.83, 2.59) | 0.02 (-2.67, 2.72) |
| Q3 | 0.38 (-2.41, 3.17) | 0.07 (-2.67, 2.81) | 0.29 (-2.40, 2.98) | 0.08 (-2.65, 2.82) | 0.30 (-2.38, 3.00) |
| Q4 (most disadvantaged) | 0.85 (-2.31, 4.02) | 0.70 (-2.40, 3.80) | 0.90 (-2.20, 4.00) | 0.77 (-2.33, 3.88) | 0.84 (-2.27, 3.95) |

[a]. Multilevel linear regression coefficients and 95% confidence intervals. Modelling adjusted for participant clustering within SA1 units.

[b]. Model 1: Neighbourhood disadvantage and waist circumference adjusting for age, sex, country of birth, education, occupation, and household income

[c]. Model 2: Model 1 plus adjustment for residential distance to the CBD (measured as the Euclidean distance between the centroid of each participant's area of residence and the capital city Central Business District)

[d]. Model 3: Model 2 plus adjustment for count of supermarkets within a 1000m buffer of the participants place of residence

[e]. Model 4: Model 2 plus adjustment for count of walkable amenities within a 1000m buffer of the participants place of residence

[f]. Model 5: Model 2 plus adjustment for count of supermarkets and walkable amenities within a 1000m buffer of the participants place of residence

There was no association between neighbourhood disadvantage and body size in Hobart, Darwin, and Canberra. These cities are the least populous of Australia's state and territory capital cities [53], and had the smallest spatial footprint as measured by the Euclidean distance of the suburbs from city CBDs. This suggests that cities may need to reach a certain critical area-size and/or population mass for concentrations of neighbourhood disadvantage to emerge (e.g., due to displacement processes such as gentrification), and hence for neighbourhood-related inequities in body size to become detectable. Additionally, in Canberra (and possibly elsewhere) urban planning and design has created small pockets of disadvantage (e.g., public housing) within larger more advantaged areas, thus obscuring the true extent of spatial inequity across the city [54] and mitigating against finding associations between neighbourhood disadvantage and body size.

In the current study, obesogenic environments were conceptualised and measured based on residential distance to the CBD, and the availability of supermarkets and walkable amenities. To contribute to the association between neighbourhood disadvantage and body size these factors needed to be related to both. In most capital cities we found the least disadvantaged neighbourhoods were closer to the CBD than more disadvantaged neighbourhoods which tended to be located furthest from the CBD. Similar findings have been reported in other Australian studies [55]. We also found residents of neighbourhoods more distant from the CBD had greater BMI (in seven of eight cities [Darwin is the exception]) and a larger waist circumference (in five cities, with Hobart, Darwin and Canberra the exceptions) than residents of neighbourhoods closer to the CBD, which also concurs with previous Australian research [44]. Building on the foregoing, our multivariable modelling showed that when associations between neighbourhood disadvantage and body size were adjusted for residential distance to the CBD, the magnitude of the neighbourhood differences were attenuated in Sydney, Melbourne, Brisbane, Adelaide, and Perth: between 28.5% and 4.0% for BMI, and between 26.8% and 4.7% for waist circumference. These findings reflect observations of previous Australian studies showing residents of socioeconomically advantaged neighbourhoods located close to the CBD are more likely to be exposed to environments conducive to a healthier bodyweight [37]. This includes living in compact neighbourhoods with interconnected street networks and diverse land use, providing shorter and more direct travel routes, thus increasing the propensity to use active travel (walking and cycling) and public transport, and reducing reliance on private motor vehicles. In short, it seems that environments which make a city liveable and healthy are disproportionately the preserve of those who can afford to reside and work near the CBD [43].

Supermarket counts varied significantly between the quintiles of neighbourhood disadvantage for all capital cities except Brisbane. In Sydney, Adelaide, Hobart, Darwin, and Canberra, supermarket density was lowest in the least disadvantaged quintile, but across the other quintiles there was no clear pattern of association. These findings do not provide compelling evidence that disadvantaged neighbourhoods in Australia's capital cities have fewer local supermarkets and less opportunity to purchase healthy food. This contrasts with the findings of a recent systematic review of socioeconomic inequities in the food retail environment in Australia [18]. However, the review used a more comprehensive definition of the food environment including retail outlets beyond just supermarkets (e.g., fast food stores, greengrocers) and factors such as access and affordability, which may account for the contrasting conclusions. Supermarket count was negatively associated with body size in Sydney, Melbourne, Brisbane, and Hobart, which is consistent with previous Australian studies [45]. That supermarket count was not related to body size in Adelaide, Perth, Darwin, and Canberra, confirms associations observed in each capital city are not necessarily generalisable to other cities. We subsequently examined to what extent supermarket count contributed to associations between

neighbourhood disadvantage and body size after adjusting for residential distance to the CBD. This was done to offset the potentially confounding effects of unmeasured associations between proximity to the CBD and supermarket availability (i.e., fewer supermarkets further from the CBD), as has been reported in previous Australian research [56, 57]. Our study found little evidence to support a conclusion that differences between advantaged and disadvantaged neighbourhoods in supermarket density contribute meaningfully to neighbourhood inequities in body size.

Counts of walkable amenities varied significantly between advantaged and disadvantaged neighbourhoods in all capital cities except Perth, although the pattern of association was not clearly interpretable. In Sydney, Hobart and Darwin, the least disadvantaged neighbourhoods had the smallest number of walkable amenities, in Melbourne the least disadvantaged had the largest number of amenities, and in Brisbane, Adelaide, Perth and Canberra, associations between disadvantage and amenities were indeterminate. Count of local walkable amenities was negatively associated with BMI in Sydney, Brisbane, Perth, and Canberra, and with waist circumference in Sydney, Brisbane, and Canberra: in these cities, residents of neighbourhoods with more walkable amenities had healthier body sizes than residents of neighbourhoods with fewer amenities. Similar results have been reported elsewhere [58–60]. Despite this evidence, our multivariable analysis found walkable amenities did not contribute to differences between advantaged and disadvantaged neighbourhoods in body size. This is unsurprising given that associations between disadvantage and amenities showed a mixed and inconsistent pattern.

There are several methodological strengths and limitations relevant to interpreting and understanding our study findings. The 2017–18 NHS collected data from all Australian states and territories, enabling city-specific analysis and improving study generalisability relative to earlier Australian research on disadvantage, the built environment, and body size, which has been largely restricted to specific cities, regions, or states [61]. The outcome measures, BMI and waist circumference, were measured rather than self-reported, an improvement over most previous Australian and overseas studies which relied on self-reported body size which is influenced by social desirability bias [62]. Where NHS participants did not consent to being measured, body size data were imputed by the ABS based on self-reported body size and other information, and these derived estimates closely reflected age- and sex-matched body-size values produced by the objective measurement. The NHS achieved a relatively high response rate of 76.1%; however, based on previous research [63, 64], it is likely that the 23.9% non-response disproportionately underrepresented people from socioeconomically disadvantaged areas and backgrounds, thus our findings may underestimate the true magnitude of the relationships examined. Given the cross-sectional nature of the NHS, assumptions about causality need to be made with caution: longitudinal research examining disadvantage, the built environment, and body size would provide stronger evidence for causal inference. We used 'residential distance to the CBD' as a broad proxy measure for obesogenic built environments, justifying our decision on the basis of extant Australian research which has shown that neighbourhoods close to and distal from a city's CBD have better and worse access respectively to environments that are conducive to healthy bodyweight. The use of specific measures of an obesogenic built environment would have been preferred, however, such measures were not available in the NHS. Despite this limitation, residential distance to the CBD was significantly associated with neighbourhood disadvantage and it accounted for some of the variation between advantaged and disadvantaged neighbourhoods in BMI and waist circumference, which suggested that relative proximity to the CBD was capturing some unmeasured aspects of an obesogenic environment. Lastly, neighbourhoods were grouped into quartiles to reflect different average levels of area-level disadvantage: this categorisation may have resulted in a loss of sensitivity, as there may be neighbourhoods within the most disadvantaged quartile that diverged from the average and

experienced disproportionate disadvantage and limited availability of supermarkets and walkable amenities, with consequent negative impacts on body size and subsequent chronic disease.

## Conclusions

Variation between the eight capital cities in the existence and magnitude of neighbourhood socioeconomic differences in body size suggests that what is observed in one city is not necessarily generalisable to all other cities. Living in socioeconomically disadvantaged neighbourhoods in most (but not all) of Australia's capital cities provides greater risk of unhealthy body size. Given the strong relationship between body size and chronic disease, neighbourhood inequities in body size likely underlies concomitant inequities in rates of morbidity and mortality for CVD, T2D and other chronic conditions. Part of the association between disadvantage and body size appears due to differences between advantaged and disadvantaged neighbourhoods in their urban form (i.e., liveability) as captured by proximity to a capital city's CBD, although the specific factors driving this association have yet to be identified and should form the focus of future research. The contribution of supermarkets and walkable amenities to the association between disadvantage and body size was indeterminant based on the findings of this study. At a minimum, the fact that disadvantaged neighbourhoods in Australia's capital cities appeared no worse off than advantaged areas in terms of the availability of supermarkets and amenities points to the potentially obesity-protective effect of living in neighbourhoods with local shops selling healthy food and having walkable access to amenities and destinations. Taken together, the results of this study point to the important role of urban design and city planning as mechanisms for addressing social and economic inequities in Australia's capital cities, and as solutions to this country's overweight and obesity epidemic and associated rising rates of chronic disease.

## Acknowledgments

The authors acknowledge the Australian Bureau of Statistics for their assistance with access to the data analysis environment. We particularly acknowledge Cassandra Elliott, Assistant Director Health National Statistics Centre at the Australian Bureau of Statistics. Analyses presented in this paper used Australian Bureau of Statistics National Health Survey Data. The views expressed in this paper are those of the authors, and do not necessarily reflect those of the Australian Bureau of Statistics. The authors also acknowledge Associate Professor Neil Coffee for his contributions to developing this research project.

## Author Contributions

**Conceptualization:** Suzanne J. Carroll, Gavin Turrell.

**Formal analysis:** Gavin Turrell.

**Methodology:** Suzanne J. Carroll, Michael J. Dale, Gavin Turrell.

**Visualization:** Michael J. Dale.

**Writing – original draft:** Suzanne J. Carroll, Michael J. Dale, Gavin Turrell.

**Writing – review & editing:** Suzanne J. Carroll, Michael J. Dale, Gavin Turrell.

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
