## [Decision Letter · Decision Letter 0]

30 Aug 2022

PONE-D-22-07257

Neighbourhood socioeconomic disadvantage and body size in Australia’s capital cities: the contribution of obesogenic environments

PLOS ONE

Dear Dr. Carroll,

Thank you for submitting your manuscript to PLOS ONE. After careful consideration, we feel that it has merit but does not fully meet PLOS ONE’s publication criteria as it currently stands. Therefore, we invite you to submit a revised version of the manuscript that addresses the points raised during the review process.

We look forward to receiving your revised manuscript.

Kind regards,

Sungwoo Lim, DrPH

Academic Editor

PLOS ONE

https://journals.plos.org/plosone/s/file?id=ba62/PLOSOne_formatting_sample_title_authors_affiliations.pdf".

Reviewers' comments:

Reviewer's Responses to Questions

**Comments to the Author**

1. Is the manuscript technically sound, and do the data support the conclusions?

Reviewer #1: Yes

2. Has the statistical analysis been performed appropriately and rigorously? 

Reviewer #1: No

3. Have the authors made all data underlying the findings in their manuscript fully available?

Reviewer #1: Yes

4. Is the manuscript presented in an intelligible fashion and written in standard English?

Reviewer #1: Yes

5. Review Comments to the Author

Reviewer #1: This paper analyzed the association between neighborhood disadvantage and body size, as well as the contribution of obesogenic environments to this association, and compared the associations across 8 cities. This paper is well-written and is an important addition to the literature. However, there are a few concerns that prevent me to endorse its acceptance at the present stage.

Introduction.

Page 5 opening paragraph. I think it is not appropriate to use “generalizability” in aims. For my understanding, study results can be generalized to a broader group of individuals with same characteristics. However, populations from two cities are geographically different, so study results from city A cannot be generalized to city B. Authors can think of a better word. Or just simply say to compare associations across cities.

Methods.

Page 6 line 117-118: It is not so clear to me how the contextual variables were collected e.g. count of supermarkets and count of walkable amenities). Were GIS methods used?

Page 6. DAG. It is a little strange to include paths that are not tested in this study. Authors can think of highlighting the paths that are the focus of the present study.

It is not clear why authors used residential location as an indicator of obesogenic environment. Should cite some papers to support. In addition, density of/ access to fast food restaurants were widely used to measure obesogenic environment. Have authors considered using that?

Page 7 line 127: individual factors influence neighborhood selection. It is possible that the association of disadvantaged neighborhoods and body size is spurious, i.e., the association is caused by a third “confounding” factor: individual-level socioeconomic status. How do authors rule out this possibility?

Analytic Approach

I don’t’ understand why authors did not combine samples of 8 places and use the full sample to test 1) the association between neighborhood disadvantage and body size, and 2) contribution of obesogenic environments to this association. For these aims, I did not see a necessity of separating participants of 8 places.

The way of testing mediation effect of obesogenic environments did not tell you whether a mediation effect is significant or not. Sequence equation modeling (SEM) or generalized sequence equation modeling (GSEM) can do a greater job by testing the significance of mediator. Recommend authors consider SEM.

6. PLOS authors have the option to publish the peer review history of their article (what does this mean?). If published, this will include your full peer review and any attached files.

Reviewer #1: No

---

## [Author Response · Author response to Decision Letter 0]

7 Oct 2022

Please see attached 'revise and resubmit' document for detailed responses

---

## [Decision Letter · Decision Letter 1]

17 Nov 2022

PONE-D-22-07257R1Neighbourhood socioeconomic disadvantage and body size in Australia’s capital cities: the contribution of obesogenic environmentsPLOS ONE

Dear Dr. Carroll,

Thank you for submitting your manuscript to PLOS ONE. After careful consideration, we feel that it has merit but does not fully meet PLOS ONE’s publication criteria as it currently stands. Therefore, we invite you to submit a revised version of the manuscript that addresses the points raised during the review process.

We look forward to receiving your revised manuscript.

Kind regards,

Sungwoo Lim, DrPH

Academic Editor

PLOS ONE

Reviewers' comments:

Reviewer's Responses to Questions

**Comments to the Author**

1. If the authors have adequately addressed your comments raised in a previous round of review and you feel that this manuscript is now acceptable for publication, you may indicate that here to bypass the “Comments to the Author” section, enter your conflict of interest statement in the “Confidential to Editor” section, and submit your "Accept" recommendation.

Reviewer #2: (No Response)

2. Is the manuscript technically sound, and do the data support the conclusions?

Reviewer #2: Partly

3. Has the statistical analysis been performed appropriately and rigorously? 

Reviewer #2: I Don't Know

4. Have the authors made all data underlying the findings in their manuscript fully available?

Reviewer #2: No

5. Is the manuscript presented in an intelligible fashion and written in standard English?

Reviewer #2: Yes

6. Review Comments to the Author

Reviewer #2: This paper seeks to examine the contribution of obesogenic environments to the association between neighbourhood disadvantage and body size. While the paper seems generally well written, I have a few comments/ suggestions that the authors may wish to consider.

1. Pg 13: The discussion of methods should include explicit mention of multilevel modelling approach, with the necessary details about units of analysis/ levels included. Currently, it isn’t explicitly clear what the authors mean by ‘accounting for clustering of individuals within SA1s”, and the term multilevel only appears subsequently in the paper.

2. In the introduction, the authors highlight that they were “documenting the association between neighbourhood disadvantage and body size and examining the contribution of obesogenic environments to this association.” The choice of word ‘contribution’ here suggests that the authors are looking at how obesogenic environment are along the causal pathway between neighborhood disadvantage and body size—i.e. how it acts as a mediator. The DAG also suggests that the authors are indeed treating the obesogenic environmental characteristics as mediators as they are visually represented as being part of the causal pathway.

Yet at the same time, some of the discussion within the paper seems to suggest that the authors are seeing obesogenic environments are potential confounders instead. For instance, in the discussion about why distance from CBD might relate to neighborhood disadvantage and BMI/waist circumference (pg 10 tracked changes version), the authors seem to be suggesting that other factors related to distance to CBD (e.g. urban sprawl, public transport usage) which are co-occurring with neighborhood disadvantage BUT not directly attributable/ related to it might be driving the association. In other words, they seem to be describing ‘distance to CBD’ as a confounder.

Are the authors looking at obesogenic environments as potential mediators or confounders of the relationship between neighborhood disadvantage and body size? This distinction should be made clear up front and be consistently applied throughout the paper, as this would affect how their findings should be interpreted.

3. I find the choice of term ‘residential location’ rather unintuitive. Why not just call it as it is, ‘distance to CBD’ ?

4. Additionally I am not entirely clear whether ‘residential location’ as defined in this paper is meant to be an ‘obesogenic environment’ characteristics being tested, or more of a control variable. My read is that the authors are treating it as the former. If so, I question the value of having such a broad ‘obesogenic environmental characteristic’ which is actually more of a proxy for many other more specific characteristics that the authors themselves highlight (pg 10 tracked changes version, where they discuss urban sprawl; use of public transport; other facets of walkability that are correlated with distance to CBD) that might explain why distance to CBD might be related to neighborhood disadvantage as well as obesity. As a ‘catch-all’ measure, it is unable to elucidate which aspects of the environment is actually obesogenic and thus needs to be addressed through urban design or policy…since ‘distance to CBD’ is not a environmental characteristic that can be changed! I would question the value of treating ‘distance to CBD’ as an obesogenic environmental characteristic.

However, if the measure is just a control for all these possible characteristics OTHER than the ones being tested (i.e. supermarkets accessibility and amenities accessibility), then I see the purpose of having this variable. If it is meant to be more of a ‘control’ variable, then the paper should be revised to clearly explain this.

5. Could the authors include a better synthesis of the results from the models looking at BMI versus waist circumference as a measure of adiposity? I found it hard to follow the current discussion—could it be clearer where results differ / concur between these two measures?

6. The authors argue that a major contribution of this paper is in its testing of the relationship of interest across different cities. To this end, I think the paper needs more introduction to the cities being studied ,especially for readers unfamiliar with Australia. Furthermore, is there a working hypothesis upfront why certain cities may exhibit certain patterns versus others? This is discussed briefly in the ‘Discussion’ section, but could the authors consider adding some in the Introduction to give clearer grounding for this particular piece of the analysis.

7. What is an ‘SA1’? What size area does it cover and what’s its population size?

7. PLOS authors have the option to publish the peer review history of their article (what does this mean?). If published, this will include your full peer review and any attached files.

Reviewer #2: No

---

## [Author Response · Author response to Decision Letter 1]

15 Dec 2022

Please see attached response to reviewers document

---

## [Decision Letter · Decision Letter 2]

26 Dec 2022

Neighbourhood socioeconomic disadvantage and body size in Australia’s capital cities: the contribution of obesogenic environments

PONE-D-22-07257R2

Dear Dr. Carroll,

We’re pleased to inform you that your manuscript has been judged scientifically suitable for publication and will be formally accepted for publication once it meets all outstanding technical requirements.

Kind regards,

Sungwoo Lim, DrPH

Academic Editor

PLOS ONE

Additional Editor Comments (optional):

Reviewers' comments:

Reviewer's Responses to Questions

**Comments to the Author**

1. If the authors have adequately addressed your comments raised in a previous round of review and you feel that this manuscript is now acceptable for publication, you may indicate that here to bypass the “Comments to the Author” section, enter your conflict of interest statement in the “Confidential to Editor” section, and submit your "Accept" recommendation.

Reviewer #2: All comments have been addressed

2. Is the manuscript technically sound, and do the data support the conclusions?

Reviewer #2: Yes

3. Has the statistical analysis been performed appropriately and rigorously? 

Reviewer #2: Yes

4. Have the authors made all data underlying the findings in their manuscript fully available?

Reviewer #2: No

5. Is the manuscript presented in an intelligible fashion and written in standard English?

Reviewer #2: Yes

6. Review Comments to the Author

Reviewer #2: I thank the authors for addressing my comments and making the edits. I have no further comments to share.

7. PLOS authors have the option to publish the peer review history of their article (what does this mean?). If published, this will include your full peer review and any attached files.

Reviewer #2: No

---

## [Editor Report · Acceptance letter]

10 Jan 2023

PONE-D-22-07257R2 

Neighbourhood socioeconomic disadvantage and body size in Australia’s capital cities: the contribution of obesogenic environments 

Dear Dr. Carroll:

I'm pleased to inform you that your manuscript has been deemed suitable for publication in PLOS ONE. Congratulations! Your manuscript is now with our production department. 

Kind regards, 

on behalf of

Dr. Sungwoo Lim 

Academic Editor

PLOS ONE